# An Approach to Intersectionally Target Mature Enteroendocrine Cells in the Small Intestine of Mice

**DOI:** 10.3390/cells13010102

**Published:** 2024-01-04

**Authors:** Christian Vossen, Patricia Schmidt, Claudia Maria Wunderlich, Melanie Joyce Mittenbühler, Claas Tapken, Peter Wienand, Paul Nicolas Mirabella, Leonie Cabot, Anna-Lena Schumacher, Kat Folz-Donahue, Christian Kukat, Ingo Voigt, Jens C. Brüning, Henning Fenselau, F. Thomas Wunderlich

**Affiliations:** 1Obesity and Cancer Research Group, Max Planck Institute for Metabolism Research, Gleueler Strasse 50, 50931 Cologne, Germany; 2Policlinic for Endocrinology, Diabetes, and Preventive Medicine (PEDP), University Hospital Cologne, 50924 Cologne, Germany; paul.mirabella@sf.mpg.de (P.N.M.); bruening@sf.mpg.de (J.C.B.); henning.fenselau@sf.mpg.de (H.F.); 3Excellence Cluster on Cellular Stress Responses in Aging Associated Diseases (CECAD), University of Cologne, 50931 Cologne, Germany; 4Center of Molecular Medicine Cologne (CMMC), University of Cologne, 50931 Cologne, Germany; 5Research Group Synaptic Transmission in Energy Homeostasis, Max Planck Institute for Metabolism Research, 50931 Cologne, Germany; 6FACS & Imaging Core Facility, Max Planck Institute for Biology of Ageing, Joseph-Stelzmann-Str. 9b, 50931 Cologne, Germany; lschumacher@age.mpg.de (A.-L.S.);; 7Transgenic Core Facility, Max Planck Institute for Biology of Ageing, Joseph-Stelzmann-Str. 9b, 50931 Cologne, Germany; ingo.voigt@age.mpg.de; 8Department of neuronal Control of Metabolism, Max Planck Institute for Metabolism Research, 50931 Cologne, Germany

**Keywords:** Dre/rox, Cre/loxP, enteroendocrine cells of small intestine, EEC, *Cck*-expressing I cell, *Sst*-expressing D cell, *Gcg*-expressing L cell

## Abstract

Enteroendocrine cells (EECs) constitute only a small proportion of *Villin-1 (Vil1)*-expressing intestinal epithelial cells (IECs) of the gastrointestinal tract; yet, in sum, they build the largest endocrine organ of the body, with each of them storing and releasing a distinct set of peptides for the control of feeding behavior, glucose metabolism, and gastrointestinal motility. Like all IEC types, EECs are continuously renewed from intestinal stem cells in the crypt base and terminally differentiate into mature subtypes while moving up the crypt–villus axis. Interestingly, EECs adjust their hormonal secretion according to their migration state as EECs receive altering differentiation signals along the crypt–villus axis and thus undergo functional readaptation. Cell-specific targeting of mature EEC subtypes by specific promoters is challenging because the expression of EEC-derived peptides and their precursors is not limited to EECs but are also found in other organs, such as the brain (e.g., *Cck* and *Sst*) as well as in the pancreas (e.g., *Sst* and *Gcg*). Here, we describe an intersectional genetic approach that enables cell type-specific targeting of functionally distinct EEC subtypes by combining a newly generated Dre-recombinase expressing mouse line (*Vil1-2A-DD-Dre*) with multiple existing Cre-recombinase mice and mouse strains with rox and loxP sites flanked stop cassettes for transgene expression. We found that transgene expression in triple-transgenic mice is highly specific in I but not D and L cells in the terminal villi of the small intestine. The targeting of EECs only in terminal villi is due to the integration of a defective 2A separating peptide that, combined with low EEC intrinsic *Vil1* expression, restricts our *Vil1-2A-DD-Dre* mouse line and the intersectional genetic approach described here only applicable for the investigation of mature EEC subpopulations.

## 1. Introduction

The intestinal epithelium is an extremely dynamic and regenerative organ. Crypt-based *Lgr5*-positive stem cells give rise to all different IEC populations along the crypt–villus axis, such as enterocytes, goblet, and tuft cells, as well as to all different enteroendocrine cells (EECs) [1]. The intestinal epithelium is completely renewed every 3–5 days [2]. Thus, most intestinal cell types differentiate and mature while migrating from the crypt towards the villus tip, where IECs die and are shed into the gut lumen. The low abundant EECs, constituting approximately 1% of the intestinal epithelium, also jump onto this conveyor belt while maturing, although EEC subtypes can be retained close to the crypt–villus junction already secreting hormones [3,4]. EECs regulate fundamental processes such as intestinal motility, mucosal immunity, glucose metabolism, and appetite via the actions of hormones that are specific to each EEC subtype [5,6]. Classic EEC annotation defined L cells (*Glp1*), I cells (*Cck*), D cells (*Sst*), EC cells (5-HT), S cells (*Sct*), X cells (*Ghrl*), K cells (*Gip*), and N cells (*Nts*) according to the expression of a specific hormone/neuroactive peptide the cells secrete (see parentheses). However, the emerging use of single-cell transcriptomics revealed that categorizing EECs might be even more complex, as the simultaneous expression of multiple, similar hormones in a coordinated manner was observed [3,7,8,9,10]. Hence, EECs can be classified into more than 20 different subtypes [5]. All EECs were found to express molecular markers such as *Neurog3* and *Neurod1* at specific developmental stages [3,11]. Therefore, making use of *Neurog3* and *Neurod1* promoters to drive the expression of Cre recombinase is amenable to the conditional manipulation of all EECs [12,13]. However, the specific manipulation of L, I, and D cells via the expression of a Cre recombinase under the control of their specific hormonal promoter is limited due to the fact that these hormones are also expressed by neuronal cell types in the brain, islet cells in the pancreas, and other sites. A significant step forward in specifically manipulating subpopulations of cells has been made via establishing the intersectional approach where two recombinase systems (either Cre/loxP and Dre/rox or Cre/loxP and Flp/Frt) are combined to define and manipulate cell types more stringently [14,15]. Two recent reports described the specific targeting of EEC subtypes via a combination of two different novel *Vil1*-*Flp*-expressing mice, restricting *Flp* expression to the intestine with existing Cre mice under the control of EEC hormonal promoters [7,10]. Using the intersectional Cre/Flp approach, both reports were able to effectively target EEC subpopulations to give important insights into how specific EEC subtypes regulate food intake, gut motility, and appetite [7,10]. However, neither report was able to address a functional distinction of villus EECs versus their crypt-based lineage precursors since *Vil1*-*Flp* transgenes were expressed constitutively in intestinal stem cells. Differences in gene expression profiles between early and mature EECs have been addressed by scRNAseq in mice expressing a bi-fluorescent protein that changes color over time [3]. The differential expressional profiles between early and late EECs revealed a time-dependent hormonal expression that highlights the plasticity during EEC development. In line with this evidence, a bone morphogenetic protein (BMP) signaling gradient along the crypt–villus axis drives the EEC flexible hormonal profile [16], indicating that villus EECs switch their hormone expression upon leaving the crypt, thus suggesting functional differences between crypt and villus EECs dependent on their location and developmental stage. However, the functional adaptations between crypt and villus EECs have not been addressed. Here, we describe an approach to specifically target differentiated villus EEC subpopulations via an intersectional Cre/Dre approach. We successfully established a temporally controllable Dre by the N-terminal fusion of a mutant *E. coli* dihydrofolate reductase (ecDHFR) to Dre in vitro. Of note, ecDHFR used as a destabilizing domain (DD) to Cre has already been shown to enable the effective temporal control of Cre-mediated recombination [17]. Thus, we generated a *Vil1-2A-DD-Dre* CRISPR/Cas9 knock-in mouse strain. We found that inducing Dre activity by trimethoprim (TMP) administration in vivo was ineffective due to the usage of a non-cleavable 2A peptide separating *Villin-1* from *DD-Dre*. Indeed, an aberrant VIL1::DD-Dre fusion protein was constantly degraded, independent of TMP. Of great interest, however, the use of our *Vil1-2A-DD-Dre* mouse line allowed specific targeting of mature IECs in the villus, and, importantly, *Vil1* inactivation did not affect intestinal physiology and morphology, consistent with previous findings [18]. We found that villus I cells, but not L and D cells, show low *Vil1* expression, as revealed by scRNAseq and RNAscope. Taken together, our newly developed genetic approach reveals critical aspects of functionally distinct EECs and constitutes a resource for targeting mature villus EEC subtypes.

## 2. Material and Methods

### 2.1. Animal Care

Mouse housing and handling were in compliance with the animal protection guidelines given by the local government authorities (Bezirksregierung Köln). All performed procedures were approved and authorized by the “Landesamt für Natur, Umwelt und Verbraucherschutz Nordrhein-Westfalen” (81-02.04.2019.A142).

Transgenic mouse lines such as *R26-rx-ZsGreen* [14], *R26-rx-fl-tdTomato* (JAX#021876), *R26-fl-tdTomato* [19], *Gcg-Cre* (JAX#030663), *Sst-Cre* (JAX#013044), *Cck-Cre* (JAX#012706), and control mice were maintained in a barrier-protected facility in individually ventilated cages (IVCs) at 22 °C and subjected to a 12 h day/night cycle. The housing conditions were improved by enrichment materials in the cage. Animals had ad libitum access to filtered water and food and were fed a normal chow diet (NCD) consisting of 57 kJ% carbohydrates, 34 kJ% proteins, and 9 kJ% fat (Ssniff). All experimental mice were sacrificed by cervical dislocation for organ collection.

### 2.2. Generation of Vil1-2A-DD-Dre Mice

*Vil1-2A-DD-Dre* mouse line was generated under license 84-02.04.2015.A025 in the Transgenesis Core Facility of MPI for Biology of Ageing using standard CRISPR/Cas9 protocols. Briefly, two gRNAs flanking the stop codon of *Vil1* were designed (g1: GTCCACCGAAGACTTCACTA; g2: CAGGACTAAAACCAGTAATG) ordered as crRNA assembled with trRNA, mixed with Cas9 protein and ssMegamer repair template (GCCACATCTGTATGATTCATCTTGAGCAAACCCAGGGGAAGACTGTGACCTTTGCCCTCTGGTCTCTTCTAGGAGCACCTGTCCACCGAAGACTTCACTAGagCCTTGGGCATGACTCCAGCTGCCTTCTCTGCCCTGCCTCGATGGAAGCAACAAAACATCAAGAAAGAAAAAGGACTGTTTGCCaccatgggtCGaGCaGAAGGaCGaGGaAGtCTtCTtAcaTGtGGaGATGTtGAAGAAAACCCAGGaCCtGGaATCTCTCTtATTGCtGCTCTtGCtGTtGACTACGTGATCGGGATGGAAAACGCTATGCCATGGAATCTGCCCGCCGATCTGGCTTGGTTCAAGAGGAACACCCTGAACAAGCCAGTGATCATGGGCAGACACACTTGGGAGTCCATTGGCCGGCCCCTGCCTGGACGCAAGAACATCATTCTGAGCTCCCAGCCCTCTACCGACGACAGGGTGACATGGGTGAAAAGTGTGGACGAAGCCATTGCCGCTTGCGGAGATGTGCCCGAGATCATGGTCATCGGCGGAGGGAGAGTGATCGAGCAGTTCCTGCCTAAGGCCCAGAAACTGTACCTGACTCACATTGACGCTGAGGTGGAAGGGGACACCCATTTTCCTGATTATGAGCCAGACGATTGGGAAAGCGTGTTCTCCGAGTTTCACGACGCCGATGCTCAGAATTCTCATAGTTATTGCTTTGAGATCCTGGAAAGGAGAGGCGCGCCtcTTaagaagaagaggaaggtgggtgctagcgagctgatcatctctggctcctctggaggattcctgaggaacatcggcaaggagtaccaggaggctgctgagaacttcatgagattcatgaatgaccagggagcctacgcccctaacaccctgagagacctgaggctggtgttccactcctgggctagatggtgccacgctagacagctggcctggttccctatctctcctgagatggctagggagtacttccttcagctgcacgatgctgacctggcctctaccaccatcgacaagcactacgccatgctgaacatgctgctgtcccactgtggcctgcctcctctgtctgatgacaagtctgtgagcctggccatgaggagaatccggagagaggctgccaccgagaagggagagagaaccggccaggccatccctctgagatgggatgacctgaagctgctggatgtgctgctgtctagatctgagagactggtggacctgaggaatagggccttcctgtttgtggcctacaacaccctgatgaggatgtctgagatctctaggatcagagtgggagacctggaccagaccggagacaccgtgaccctgcacatctcccacaccaagaccatcaccaccgctgctggcctggacaaagtgctgtctagaaggacaaccgctgtgcttaatgactggcttgatgtgtctggtcttagagagcatccggatgctgtgctgttccctcctatccaccggagcaacaaggctaggatcaccaccacccctctgaccgcccctgccatggagaagatttttagcgatgcctgggtgctgctgaacaagagggatgccacccctaacaagggccgctaccggacctggaccggccactctgctagagtgggagctgccatcgacatggctgagaagcaagtgtccatggtggagatcatgcaggagggcacctggaaaaagcctgagacactgatgagatacctgaggaggggaggagtgtctgtgggagccaactctaggctgatggactccgctagcggcgcctgaattcCAGATGAAAATGTCCAATTGTACCTGTGAGCCACAGTGTGACAATTCCTTTTGTTTATAATAGTAATTTGCCCATTCCTTCAGACGCATGCCACAGACCCATGG) to inject into C57BL/6N oocytes. Injected oocytes were implanted into oviduct of foster mothers; 15 offspring were born, from which 5 carried the *Vil1-2A-DD-Dre* insertion verified by PCR using primers (5′Vil1typ: GTGGCCTGACTGACTACACGGACA; 3′Vil1typ: CCAGAGAGCTTCAATTCTCAAAA; 3′Dre: CCTCAGGAATCCTCCAGAGGA) to result in 819 bp targeted and 310 bp wt bands. R1 and R2 PCR reactions described in Figure 2 were carried out for founder 4. Bands were subcloned into pGEM-T easy (Promega) and sequenced (Appendix A).

### 2.3. In Vivo DD-Dre Recombination Activity Assessment

Intestinal Vil1-DD-Dre efficiency was analyzed in *R26-rx-STOP-rx-ZsGreen* reporter mice carrying the *Vil1-2A-DD-Dre* allele either hetero- or homozygously. A total of 300 µg/g BW TMP solution (Trimethoprim lactate salt, Sigma (Taufkirchen, Germany) T0667 in 20% DMSO in 1× PBS) was administered to adult female and male mice via i.p. injection or oral gavage on three consecutive days. Mice were sacrificed 1–3 days after the last injection, and the small intestine and colon were removed. The small intestine was divided into three parts representing the duodenum, jejunum, and ileum from proximal to distal. Intestinal parts were opened longitudinally, and intestinal contents were removed using forceps. Next, swiss roll mounts were generated, and the tissue was covered with tissue freezing medium (Leica) in cryomolds and frozen on dry ice without fixation. Moreover, 10 µm sections were generated with a cryostat and mounted onto SuperFrost Plus slides (ThermoFisher, Langenselbold, Germany). For ZsGreen quantification, slides were thawed 10 min in ice-cold 4% PFA in PBS solution. Residual PFA was washed off 3 × 5 min with PBS, and sections were embedded and counterstained using Vectashield Antifade Mounting Medium containing DAPI (Vector Laboratories, Newark, CA, USA). Sections were imaged, and ZsGreen^+^ IECs were quantified using Fiji (NIH), as stated below.

### 2.4. Analysis of Gene Expression

Isolated IECs, crypts, or intestinal pieces were homogenized in QIAzol lysis reagent (Qiagen) using a T10 Ultra-Turrax homogenizer (IKA, Stauffen, Germany). Chloroform was added, and the samples were vortexed and left for 3 min on the bench. Next, phases were separated by centrifuging for 15 min at 12,000 rpm and 4 °C. The aqueous phase was transferred and mixed with one volume of 70% EtOH. Afterward, total RNA was purified using an RNAeasy Mini Kit (#74106, Qiagen, Hilden, Germany) according to the manufacturer’s instructions. DNA was digested for 15 min on the spin columns using an RNase-Free DNase set (#79254, Quiagen). RNA was reversely transcribed into cDNA using the High-Capacity cDNA Reverse Transcription Kit (#4368813, Applied Biosystems, Waltham, MA, USA). Additionally, 50–100 ng cDNA was used for Real-time qPCR-based expression analysis using Takyon Low ROX Probe 2× MasterMix dTTP blue (#UF-LPMT-B0701, Eurogentec, Seraing, Belgium) and TaqMan probes (*Alpi*-FAM: Mm01285814_g1, *Lgr5*-FAM: Mm00438890_m1, *Tbp*-VIC: Mm00446973_m1). DD-Dre expression was determined using a custom-designed probe (*DD-Dre*-FAM probe 5′-3′: AAG GTG GGT GCT AGC GAG CTG A; Primer 5′-3′: 5′ AGA TCC TGG AAA GGA GAG GC; 3′ CTC AGG AAT CCT CCA GAG GA). Gene expression analysis was performed using a QuantStudio 7 Flex Real-Time PCR System and the QuantStudio Real-Time PCR Software v1.7.1 (Life Technologies, Carlsbad, CA, USA). Target gene expression was normalized to the mRNA expression levels of the TATA-binding protein (*Tbp*) and analyzed using the 2^−ΔΔCt^ method. Expression levels are represented relative to the respective control condition.

### 2.5. IEC Isolation and Ex Vivo Treatment

The small intestine and colon were removed from mice, and the small intestine was divided into three parts representing the duodenum, jejunum, and ileum from proximal to distal. Intestines were washed in ice-cold PBS by vortexing. Then, the samples were incubated for 30 min at 37 °C with horizontal rotation in 20 mL predigestion solution (1× HBSS without magnesium/calcium, 10 mM HEPES, 5 mM EDTA, 1 mM DTT, and 2% FCS). IECs were detached by intensive vortexing and harvested on ice using a 100 μm cell strainer (Greiner, Kremsmünster, Austria). To increase the IEC yield, samples were vortexed a second time in 10 mL predigestion solution, and cell suspensions were pooled. Afterward, IECs were collected by centrifugation (10 min at 50 g and 4 °C) and resuspended in 1X HBSS/2%FCS solution. IECs were treated with 2 µM TMP for 4 h at 37 °C with horizontal rotation. Cells were collected by centrifugation (5 min at 8000 rpm and 4 °C) and snap-frozen in liquid nitrogen.

### 2.6. Crypt–Villus Fractionation

The proximal 5 cm of the small intestine was removed from mice and opened longitudinally in ice-cold PBS. To obtain the villus fraction, villi were gently scraped off using a hemocytometer coverslip and snap-frozen in liquid nitrogen. To isolate primary crypts, the remaining tissue was cut into 1–2 mm pieces and washed in PBS by pipetting up and down with a Pasteur pipette. Then, crypts were detached by means of incubation in 2.5 mM EDTA in PBS solution for 40 min at 4 °C. Crypts were isolated by pipetting up and down in 10 mL 1% FCS in PBS using a Pasteur pipette and collected on ice using a 70 µm cell strainer. To increase the crypt yield, the last step was repeated, and fractions were pooled. Afterward, crypts were collected by centrifugation (800 rpm for 5 min at 4 °C) and snap-frozen in liquid nitrogen.

### 2.7. Immunohistochemistry

All tissues used for immunofluorescence stainings or in situ hybridization were prepared, as described in the following: the small intestine and colon were removed from mice, flushed with ice-cold PBS using a syringe, and opened longitudinally. The small intestine was divided into three parts representing the duodenum, jejunum, and ileum from proximal to distal. Swiss roll mounts were generated and fixed for 24 h in a cassette immersed in 4% PFA in PBS at 4 °C. Samples were cryoprotected in 15% (*w*/*v*) sucrose in PBS, followed by 30% (*w*/*v*) sucrose in PBS until the tissue sank. Swiss rolls were embedded in tissue-freezing medium (Leica). Moreopver, 10 µm thick sections were generated with a cryostat and mounted onto SuperFrost Plus (ThermoFisher) slides for immunostaining or SuperFrost Plus Gold slides (ThermoFisher) for in situ hybridization.

### 2.8. In Situ Hybridization (RNA Scope)

Sections were thawed, washed with PBS, and baked for 1 h at 60 °C. Afterward, slides were soft-boiled in Target Retrieval solution (Cat No. 322000, ACD) at 95–99 °C for 12 min, rinsed in autoclaved Millipore water and dehydrated for 1 min in 100% ethanol. A hydrophobic barrier was painted around the samples using an ImmEdge hydrophobic barrier pen (Vector Laboratories), and the slides were left to dry overnight in the dark. The following incubation steps were carried out at 40 °C using the HybEZ hybridization system (#321462, ACD). Sections were treated with Protease Plus (#322331, ACD) for 25 min. Next, probe hybridization and amplification were performed according to the manufacturer’s instructions (Multiplex Fluorescent Detection kit v2, #323110, ACD). Target probes were multiplexed and tyramide-conjugated to Opal520, Opal570, or Opal650 fluorophores diluted 1:750 in TSA buffer (PerkinElmer, Waltham, MA, USA):
**Target Probes**Custom-Dre-C1 (442,641)Cck-C1 (402,271)Gcg-C1 (400,601)Sst-C2/C3 (404,631)Tac1-C1 (410,351)Tdtomato-C2/C3 (317,041)ZsGreen-C2 (461,251)Vil1-C3 (463,301)

Finally, sections were counterstained with DAPI and mounted using Vectashield Antifade Mounting Medium (Vector Laboratories).

RNA in situ hybridization was combined with immunofluorescence staining in order to co-label EEC marker expression and tdTomato protein abundance. For this, in situ hybridization was performed as described above, except for protease plus treatment, which was reduced to 15 min. Further, samples were not counterstained with DAPI after the final wash of the Multiplex Fluorescent Detection kit v2 (ACD), but instead, samples were washed 2 × 5 min in PBS. Thereafter, it was continued by incubating the samples in 0.3% glycine in PBS. All following steps were performed as described for immunofluorescence.

### 2.9. Immunofluorescence

Residual tissue-freezing medium was washed off the Swiss roll sections with PBS. A hydrophobic barrier was painted around the samples using an ImmEdge hydrophobic barrier pen (Vector Laboratories). Next, samples were treated for 10 min with 0.3% glycine in PBS, washed for 10 min with PBS, and permeabilized for 10 min with 0.03% SDS in PBS solution. Unspecific binding sites were blocked for 1 h with blocking solution (1× PBS, 3% donkey serum, 0.25% TritonX). Then, the primary antibody was applied in blocking solution (Goat Polyclonal anti-tdTomato, 1:4000, Sicgen, #AB8181-200) overnight at 4 °C. The next day, slides were washed 3 × 10 min with 0.1% TritonX in PBS. The secondary antibody (Donkey Polyclonal anti-goat IgG (H + L) Alexa Fluor 647, 1:500, Invitrogen, #A-21447) and DAPI (1:1000) were applied in 0.25% TritonX in PBS for 1 h at RT, followed by washing 3 × 10 min with 0.1% TritonX in PBS. Finally, slides were mounted using Vectashield Antifade Mounting Medium (Vector Laboratories).

### 2.10. Image Acquisition and Analysis

All images were acquired with an Olympus Slideview VS200 system using a 20×/0.80 UPlanXApo objective (Olympus, Hamburg, Germany). Afterward, tiff images were imported into Fiji (NIH) for further analysis.

As measurement for intestinal Vil1-DD-Dre activity and inducibility in vivo, ZsGreen^+^ IECs were quantified across the crypt–villus axis in duodenum, jejunum, ileum, and colon. Therefore, three crypts/villi of each mouse were divided into 4 equal quadrats, of which 1/4 represented the top 25% of the crypt–villus axis (villus tip) and 4/4 represented the bottom 25% of the crypt–villus axis (the crypt). Next, ZsGreen^+^ nuclei per total IEC number (DAPI^+^ nuclei) were determined per quadrant. DD-Dre activity is represented as mean of relative ZsGreen positivity per mouse.

### 2.11. Single-Cell RNA Sequencing of the Murine Intestinal Epithelium

Intestinal epithelial cells of C57BL/6N wildtype mice were isolated from the duodenum, jejunum, and ileum, as stated above. Isolated cells from three 12-week-old male mice were pooled in order to minimize artifacts originating from individual mice. IECs and intraepithelial immune cells were resuspended in 2% FCS in PBS and counterstained using propidium iodide (PI). Live cell sorting was performed with a BD FACSAria Fusion and BD FACSAria IIIu at 4 °C using a 100 μm nozzle and sheath pressure set at 20 psi. Live single cells were gated based on PI fluorescence. Sorted, PI-negative cells were collected in 1× PBS buffer containing 0.04% BSA for 10× genomics single-cell sequencing.

Isolated cells were processed using a 10× Genomics Chromium Controller. cDNA was prepared using the Chromium single-cell 3′ reagent kit v3 according to the manufacturer’s protocol (10× Genomics). The resulting cDNA was sequenced on Illumina NovaSeq 6000 systems, generating a library with an estimated number of 8.139–9.595 cells/sample. Cell ranger analysis (10× Genomics) was performed using version 3.0.2. Unique transcript (UMI) matrices of all three samples were merged into one object and analyzed in R (2023.6.2) using Seurat v4.3.0.1 (https://doi.org/10.1016/j.cell.2019.05.031). Cells were filtered, removing cells with less than 800 or more than 50,000 UMIs, less than 200 or more than 6000 features, and more than 40% of mitochondrial genes. Further, the standard Seurat workflow was used to integrate data across the different intestinal segments. (https://github.com/satijalab/seurat/blob/release/4.3.0/vignettes/integration_introduction.Rmd, acessed on 24 october 2023). Additionally, 2000 highly variable features and 30 principal components were used for graph generation and umap calculation. FindClusters at resolution = 0.8 was used to generate clusters on the integrated dataset. Clusters were manually annotated based on the markers in Appendix A and panglaodb.se. After identifying enteroendocrine cells (EECs) based on Chga and Neurod1 expression, EECs were isolated using subset. EECs were re-clustered in the RNA assay using the standard Seurat pipeline (https://satijalab.org/seurat/articles/pbmc3k_tutorial.html, acessed on 18 August 2022) with the number of variable features to be used reduced to 200. A total of 30 principal components were used for graph generation and umap calculation, and FindClusters were run at resolution = 1.5. As we only ended up with 169 cells, unsupervised clustering was not sufficient to efficiently separate EEC subtypes (I-cells, D-cells, L-cells, K-cells, EC-cells, N-cells). Therefore, EEC subtypes were defined by their respective marker expression (I-cells = Cck > 4, D-cells = Sst > 4, L-cells = Gcg > 3, K-cells = Gip > 4, EC-cells = Tph1 > 2.5, N-cells = Nts > 4) and annotated as Cck_high, Sst_high, Gcg_high, Gip_high, Tph1_high, and Nts_high cells allowing to compare gene expression between subpopulations. In this step, 15 cells were omitted as none of the above markers exceeded the respective expression threshold. Single-cell RNA sequencing data were deposited in NCBI Gene Expression Omnibus with an identifier: GSE250433

### 2.12. Cell Culture Experiments

Mouse embryonic fibroblasts (MEFs) carrying a CAG-rox-stop-rox-ZsGreen reporter construct in the ROSA26 locus were SV40 immortalized via transfection of an SV40-expression plasmid. MEFs were cultured in Dulbecco’s Modified Eagle Medium (DMEM) GlutaMAX™ (ThermoFisher) supplemented with 10% FCS (PAN-Biotech, Aidenbach, Germany), 1% penicillin/streptomycin, 1% non-essential amino acids, and 1% sodium pyruvate (all ThermoFisher) and kept on tissue-treated culture plates (TPP^®^) at 37 °C in a 95% humidified atmosphere with 5% CO_2_. Cells were transfected with pTE-DD-Dre/pET-Dre and pCAGGS-mCherry using Lipofectamine^®^ 2000 Reagent (Thermo Fisher) according to the manufacturer’s instructions. Then, 24 h post-transfection, MEFs were treated with 2 µM TMP (Sigma-Aldrich, Taufkirchen, Germany). Fluorescence images were taken using a Leica DMI3000B widefield Microscope (Leica).

### 2.13. Flow Cytometry

Quantitative flow cytometry was performed using the MACSQuant^®^ VYB Flow Cytometer (Miltenyi Biotech, Bergisch Gladbach, Germany). Cell pellets were resuspended in FACS buffer (Miltenyi Biotec). To sort out apoptotic cells, samples were stained 1:1000 with Aqua Fluorescent Reactive Dye (Thermo Fisher). The emission of ZsGreen was measured to determine the efficiency of the DD-Dre system, while the emission of mCherry served as a transfection control. The following gating strategy was used: forward scatter-area (FSC-A) vs. side scatter-area (SSC-A) dot blots for cell identification. Single cells were gated via side scatter-height (SSC-H) vs. SSC-A and forward scatter-height (FSC-H) vs. FSC-A dot blots. Dead and apoptotic cells are gated out according to their Aqua fluorescence. Data were analyzed by MACSQuant^®^ Analyzer 10 (Miltenyi Biotech) and Flowjo (Treestar, Woodburn, OR, USA).

### 2.14. Immunoblot Analysis and Quantification

To detect protein levels, immunoblot analysis was performed. Cells were resuspended in cell lysis RIPA buffer (50 mM Tris-HCL, pH 7.5, 150 mM NaCl, 1 mM EDTA, 0.1% Natrium-Deoxycholate and phosphor-inhibitor cocktail (Roche, Basel, Switzerland)) and lysis was accelerated by three cycles of snap-freezing samples in liquid nitrogen. Protein concentration was determined with a bicinchoninic acid (BCA) assay by Pierce™ BCA Protein Assay Kit (ThermoFisher) according to manufacturer’s instructions. Cell lysates were diluted with Laemmli buffer (Bio-Rad) containing 10% β-mercaptoethanol (Applichem, Darmstadt, Germany) and boiled for 10 min at 95 °C. Proteins were separated according to the molecular weight on a gradient Protean TGX gel (Bio-Rad) at 200 V and blotted on a nitrocellulose membrane (Bio-Rad) using standard protocols. The membrane was washed three times for 10 min with 1× TBS containing 0.1% Tween (TBS-T) (Applichem) followed by a blocking step with TBS-T, containing 10% Western blocking reagent (WBR) (Roche Diagnostics) for 1 h at room temperature. Primary antibodies (polyclonal anti-dre, 1:1500, Diagenode, C15310170; anti-calnexin, 1:1000, Merck, AB2301; ant-villin-1, 1:1000, Invitrogen, PA5-22072; anti-mCherry, 1:2500, Invitrogen, AB356482) were applied in TBS-T containing 5% WBR at 4 °C overnight. On the next day, after three washing steps, anti-rabbit secondary antibody (Sigma-Aldrich, AB228341) was applied 1:4000 in TBS-T containing 5% WBR for 1 h at RT. The blot was analyzed using SuperSignal™ West Dura Luminol/Enhancer solution (Thermo Fisher). Bands were detected on a ChemiDoc MP Imaging System (Vilber Lourmat). The intensity of the bands was quantified using Fiji (NIH).

### 2.15. Statistics

Mice of indicated genotypes were assigned randomly to groups. Data are represented as mean ± SEM. Statistical significance was calculated using GraphPad prism 9. For comparison of two groups, a two-tailed unpaired Student’s *t*-test was performed. For comparison of more than two groups, a two-way ANOVA with Šídák’s multiple comparisons testing was performed. Statistical significance is indicated as follows: * *p* ≤ 0.05, ** *p* ≤ 0.01, *** *p* ≤ 0.001, **** *p* ≤ 0.0001.

## 3. Results

### 3.1. Examination of TMP-Inducible DD-Dre System In Vitro

Until now, mature EECs cannot be specifically targeted in a conditional manner. While EECs initiate hormonal gene expression in the crypt, a BMP gradient from the crypt to the villus alters the expression profiles during maturation [3,16]. The specific tracking and manipulation of these villus EECs might be possible using an intersectional and inducible approach that allows the restriction of transgene expression to villus-migrated EECs. To this end, we sought to employ intersectional genetics by using existing Cre mice with a temporally controlled intestinal Dre recombinase system. To establish an inducible Dre system, we generated a plasmid encoding a destabilizing domain (DD) N-terminally fused to Dre. The expression of the fusion protein is regulated by the CAG promoter to test the efficiency of the DD-Dre system in vitro. Without the inducer trimethoprim (TMP), DD-Dre is targeted for proteasomal degradation, whereas TMP supplementation should stabilize DD-Dre to cause the recombination of rox-flanked targets (Figure 1a). pTE-DD-Dre or pTE-Dre plasmids were co-transfected with a mCherry-encoding plasmid into immortalized MEFs carrying a targeted ROSA26 insertion of a rox-flanked STOP cassette that prevents the expression of *ZsGreen* (Figure 1a). Moreover, 24 h after transfection, cells were treated with 2 µM TMP for 24 h and then subjected to flow cytometry (Figure 1b). While TMP treatment had no effect on pTE-Dre-mediated recombination, TMP increased the number of ZsGreen^+^ cells transfected with pTE-DD-Dre two-fold compared to non-treated controls. Of note, TMP-induced DD-Dre activity was approximately 90% of Dre, whereas, without the TMP, the background activity of DD-Dre was 45% (Figure 1b). Next, we aimed to determine the DD-Dre protein dynamics and TMP dependency using Western blot analysis (Figure 1c,d). While without TMP, low amounts of DD-Dre protein could be detected, prolonged TMP treatment stabilized DD-Dre fusion protein substantially until 8 h of treatment. Conversely, when we stimulated transfected cells for 8 h and removed TMP from the medium, a decline in DD-Dre fusion protein abundance was observed between 30 min and 8 h (Figure 1e,f). Taken together, DD-Dre shows inducible and efficient recombination in vitro that can be applied for genetic manipulation of IECs in vivo.

### 3.2. CRISPR/Cas9-Mediated Insertion of DD-Dre into IEC-Specific Vil1 Gene of C57BL/6N Mice

Next, we generated IEC-specific DD-Dre mice via CRISPR/Cas9-mediated transgenesis of the *Vil1* gene in C57BL/6N oocytes. *Vil1* is expressed exclusively in IECs, and promoter constructs were successfully used previously to express Cre recombinase in the intestine [20]. Two gRNAs flanking the STOP codon of the *Vil1* gene were co-injected with Cas9 protein and an SSmegamer repair template carrying homology arms to the *Vil1* gene and an in-frame fusion of *2A-DD-Dre* into C57BL/6N oocytes (Figure 2a). Genotyping revealed that out of 15 born pubs, 5 carried the *Vil1-2A-DD-Dre* insertion (Figure 2b). Founder mouse 4 was selected to carry out PCR reactions R1 and R2 in order to sequence the *Vil1* insertion (Appendix A). The correct integration of the 2A-DD-Dre construct in-frame to the endogenous *Vil1* gene was confirmed (Figure 2a,c). Upon establishment of the *Vil1-2A-DD-Dre* line, qPCR revealed a descending *DD-Dre* expression from the duodenum to the distal colon, as previously observed for *Vil1* [20,21]. As expected, *DD-Dre* expression was not detectable or was very low in the liver, spleen, muscle, brain, lung, pancreas, and kidney, indicating the IEC-specific *DD-Dre* expression from the endogenous *Vil1* gene (Figure 2d and Appendix A). Since low DD-Dre expression in the colon might not be sufficient for efficient Dre-mediated recombination, we crossed *Vil1-2A-DD-Dre* mice to homozygosity and examined *DD-Dre* expression via qPCR (Figure 2d). This analysis revealed an allele-dependent increase in *DD-Dre* expression throughout the GI tract with a still low expression in the proximal and distal parts of the colon. To confirm these findings on the cellular level, we performed RNAscope ISH from the small intestine and colon of *Vil1-2A-DD-Dre* heterozygous mice using probes detecting *Vil1* and *Dre* mRNAs (Figure 2e). While we observed the overlapping expression of *Vil1* and *Dre* in villi of the small intestine, the expression could not be detected in the crypts of the small intestine and colon. Consistent with this, crypt and villus fractionation followed by qPCR confirmed reduced *Vil1* expression and no detectable *DD-Dre* expression in crypts, indicating that stem cells and early progenitors exhibit lower levels of *Vil1* expression (Figure 2e,f). Of note, only the most apical, lumen-oriented enterocytes of the colon expressed *Vil1* and *Dre*. Next, we isolated small intestinal IECs from control, heterozygous, and homozygous *Vil1-2A-DD-Dre* mice and exposed them to TMP ex vivo to determine the dynamics of DD-Dre protein stabilization via Western blot (Figure 2g). However, while we failed to detect DD-Dre protein, Western blot for VIL1 revealed a genotype-dependent reduction in VIL1 protein with nearly complete absence in homozygous mice. This indicated the formation of a fusion between VIL1 and DD-Dre that is proteosomally degraded. In fact, the non-functional 2A peptide used in *Vil1-2A-DD-Dre* line fails to perform self-cleavage opposite to P2A as revealed in independent sets of transient transfection experiments with *Cre-2A-ZsGreen* and *Cre-P2A-ZsGreen* encoding plasmids (Appendix A). Conclusively, these results suggest that our *Vil1-2A-DD-Dre* insertion creates a VIL1::DD-Dre fusion protein due to a non-functional 2A peptide, and this fusion protein is constantly degraded to generate *Vil1* knock-out mice. Of note, *Vil1* inactivation was reported to neither affect the murine intestinal physiology nor morphogenesis [18].

### 3.3. TMP-Independent Dre-Mediated Recombination in the Small Intestine of Vil1-2A-DD-Dre Mice

Since *Vil1* inactivation did not cause a phenotype in mice, we next sought to examine whether *Vil1-2A-DD-Dre* mice are suitable to activate transgene expression in IECs. To this end, we generated mice with a Dre-dependent ROSA26-ZsGreen reporter construct with heterozygous and homozygous *Vil1-2A-DD-Dre* insertion to determine DD-Dre activity (Figure 3a). Mice were orally treated or i.p. injected with TMP for three consecutive days. Controls did not receive TMP. Then, 1–3 days after the last TMP administration, mice were sacrificed, intestines were removed, and ZsGreen^+^ IECs were quantified. Using this protocol, we expected approx. 50% ZsGreen^+^ IECs derived from crypts in case TMP-inducible DD-Dre recombination occurred in intestinal stem cells. To provide the special resolution of the Dre activity, the crypt–villus axis was divided into four equal quadrants, of which 1/4 represents the villus tip and 4/4 represents the crypt (Figure 3b). DD-Dre activity occurred independent of TMP treatment and was similar in heterozygous and homozygous *Vil1-2A-DD-Dre* mice. Moreover, DD-Dre activity was restricted to the small intestine, following the expression gradient from the duodenum to the ileum (Figure 2d). In line, DD-Dre activity was virtually absent in the colon (Figure 3c–f). According to the expression pattern of *Vil1* and *DD-Dre*, recombination efficiency was absent in 4/4 of the crypts, where intestinal stem cells reside, and started in 3/4 to increase to 2/4, ultimately resulting in 100% recombination efficiency in 1/4. Recombination efficiency was similar between heterozygous and homozygous *Vil1-2A-DD-Dre* mice with or without TMP treatment. To determine the correlation between *DD-Dre* and *ZsGreen* expression upon Dre-mediated recombination of the rox-flanked stop cassette, we performed RNAScope ISH in non-induced *Vil1-2A-DD-Dre:R26-rx-ZsGreen* mice (Figure 3g). *DD-Dre* expression preceded *ZsGreen* expression in the villus axis most likely accumulating in the differentiated IECs before getting active. Conclusively, the *Vil1-2A-DD-Dre* can be used for non-inducible Dre-mediated recombination strategies, specifically in mature IEC subpopulations of the small intestine. Despite the non-inducibility of the *Vil1-2A-DD-Dre*, its late activity in mature IECs might still enable us to specifically target mature villus EECs using an intersectional Cre/Dre approach.

### 3.4. Intersectional Expression of tdTomato in Mature EECs

While developing in the crypt, EECs express many different peptides and only show full lineage commitment to *Cck*-, *Sst*-, and *Glp*-*1* expressing I, D, and L cells when moving upwards the villus [3,16]. Here, *Glp-1*-expressing L cells remain mainly in the crypts of the colon and ileum, whereas *Sst*-expressing D cells and *Cck*-expressing I cells also localize at more apical sites throughout the small intestine (Figure 4a). To determine the potential for an intersectional approach in mature I, D, and L cells, we intercrossed *Cck-Cre*, *Sst-Cre,* and *Gcg-Cre* mice with heterozygous *Vil1-2A-DD-Dre* and Dre/Cre-dependent *ROSA26-rx-fl-tdTomato* mice (Figure 4b). *Cck-Cre* is expressed in neurons, lungs, I cells, and other EEC subtypes. Similarly, *Sst* is expressed in pancreatic islet cells, and *Sst*-Cre effectively recombines loxP-flanked stop cassettes in various neuronal populations in the brain as well as in D cells in the GI tract (Appendix A). Likewise, *Gcg*-Cre is expressed in pancreatic alpha cells and by *Glp-1*-expressing L cells in the terminal ileum and colon, as well as in the brainstem (Appendix A). Therefore, the use of a dual-recombinase system that confers intestinal specificity is indispensable in order to specifically target intestinal EEC subtypes. As *Vil1-2A-DD-Dre* expression is restricted to villus-migrated IECs in the small intestine, an intersection is created that only a population of mature I, D, or L cells should display Cre and Dre activity (Figure 4b). Triple-transgenic (*Cck-Cre* or *Sst-Cre* or *Gcg-Cre:Vil1-2A-DD-Dre*; *R26-rx-fl-tdTomato*) and control mice were sacrificed, and Swiss roll mounts of jejunum and ileum were examined for tdTomato abundance using IHC/RNAScope ISH combination (Figure 4c and Appendix A). While no tdTomato expression was observed in any of the control samples, we found tdTomato-positive cells in triple-transgenic *Cck-Cre:Vil1-2A-DD-Dre:R26-rx-fl-tdTomato* mice but not in *Sst-Cre:Vil1-2A-DD-Dre:R26-rx-fl-tdTomato* nor in *Gcg-Cre:Vil1-2A-DD-Dre:R26-rx-fl-tdTomato* (Figure 4c and Appendix A). We had expected the failure of labeling L cells using *Vil1-2A-DD-Dre* since L cells (*Gcg-Cre*) are mainly located in the colon and reside in ileal crypts. However, failing to target D cells (*Sst-Cre*) and I cells (*Cck-Cre*) with only low efficiency was rather unexpected. An RNAscope ISH revealed the widespread distribution of D and I cells throughout the crypts and villi but without detectable tdTomato expression (Figure 4c and Appendix A). Therefore, we assumed that the Dre expression in EEC cells was not sufficient to drive Dre-mediated recombination, also confirmed by RNAscope ISH in *Vil1-2A-DD-Dre-tg+/−: R26-rx-ZsGreen* mice multiplexing specific hormonal probes with a probe against *ZsGreen* mRNA (Appendix A). Next, when we performed an RNAscope against *Vil1* in combination with *Cck*, we clearly observed that *Vil1* is much lower (if at all) expressed in EECs compared to the neighboring enterocytes (Figure 4d). To validate this finding, scRNAseq from small intestinal cells was performed to determine *Vil1* expression in IECs (Figure 4e). Small intestinal cells were annotated in 12 clusters (Appendix A). When comparing *Vil1* expression between cell types, enterocytes displayed the highest and paneth cells with the lowest *Vil1* expression of the IEC-type cells (Figure 4f). Immune cells lacked *Vil1* expression. Strikingly, the EEC cluster expressed *Vil1* only at an intermediate level (Figure 4f). To further characterize *Vil1* expression in specific EEC subtypes, the initial EEC cluster was re-clustered (Appendix A). As EEC yield during IEC isolation without specific EEC enrichment is typically limited, further analysis was performed on low EEC cell counts. Therefore, unsupervised clustering was not sufficient in order to separate between classic EEC subtypes (Appendix A). To circumvent this problem, EEC clusters were manually annotated based on threshold expression of the most prominent hormones (I cells (*Cck_high*), D cells (*Sst_high*), L cells (*Gcg_high*), K cells (*Gip_high*), EC cells (*Tph1_high*), and N cells (*Nts_high*)) (Figure 4g and Appendix A). The analysis of *Vil1* expression between EEC subtypes revealed *Cck*-expressing cells with high *Vil1* expression that were not present in *Sst* and *Gcg* clusters, indicating that, indeed, the endogenous *Vil1* expression is not sufficient in D, L, and most I cells to use an intersectional targeting approach with the *Vil1-2A-DD-Dre* mouse strain (Figure 4h, i). Strikingly, *Tph1*-expressing enterochromaffin (EC) and *Nts*-expressing N EECs showed the highest *Vil1* level, indicating that an intersectional approach with respective Cre mice and *Vil1-2A-DD-Dre* would be more convenient (Figure 4i). Interestingly, early crypt ECs express *Tac1*, which is lost in villus ECs [3]. Strikingly, while an RNAscope ISH against *Tac1* and *Vil1* confirmed this finding (e.g., lower *Tac1* expression in villus vs. crypt ECs), villus ECs that are still detectable by *Tac1* expression exhibited increased *Vil1* levels compared to their crypt counterparts. Thus, these data suggest that intersecting *Tac1*-*Cre* with *Vil1-DD-Dre* mice represents the most promising approach to target mature EC population in the small intestine.

## 4. Discussion

Here, we describe the use of a dual recombinase approach composing existing Cre mice and novel *Vil1-2A-DD-Dre* mice to intersectionally target EEC subpopulations. Enteroendocrine-specific genes, such as *Glp-1* (part of the *Gcg* gene), are also expressed by pancreatic alpha cells, thereby limiting the use of single *Gcg*-Cre mice to specifically target the L-cell population in the intestine. To circumvent the problem of gene expression from ROSA26 transgenes in cells other than EECs, transgene expression is prevented by rox- and loxP-flanked stop cassettes [14,15,22]. Thus, only cells expressing both Dre and Cre recombinases will display transgene expression. To this end, we aimed to develop an inducible intestine-specific Dre system by inserting a 2A-separated TMP-inducible DD-Dre transgene into the endogenous *Vil1* gene whose promoter had successfully been used to express Cre recombinase in all IECs in *Villin-Cre* mice [20]. We examined a TMP-inducible destabilized Dre construct in vitro and observed background recombination in the absence of TMP, most likely due to the high plasmid copy number upon transient transfections. While high background recombination could become problematic for our approach, we still expected that a single copy integration of the DD-Dre construct into the endogenous *Vil1* gene would allow for TMP-inducible Dre-mediated recombination. However, although the TMP-inducible DD-Dre system worked properly in vitro, the usage of a non-self-cleavable 2A peptide in our *Vil1-2A-DD-Dre* mice generated a VIL1::DD-Dre fusion protein in IECs that is constantly degraded, even in the presence of TMP. Thus, the *2A-DD-Dre* insertion created a *Vil1* knock-out allele that, fortunately, did not affect intestinal physiology and morphology [18]. Nevertheless, intercrosses between *Vil1-2A-DD-Dre* and Dre-dependent ZsGreen reporter mice revealed largely overlapping *Vil1*, *Dre*, and *ZsGreen* RNA expression in villi but not in crypts of the small intestine. This was observed even in the absence of TMP, indicating that constant background activity of DD-Dre resulted in the excision of the rox-flanked stop cassette in differentiated villus IECs. While these experiments [23] nicely recapitulate the co-expression of *Vil1* and *DD-Dre* from the endogenous *Vil1* promoter, the marginal expression of *Vil1* and *Dre* in intestinal stem cells, from which all IECs originate, restricts the use of *Vil1-2A-DD-Dre* mice to only manipulate mature villus IECs. This is in contrast to IEC-specific *Vil1-CreERT2* mice, where the expression of CreERT2 is driven by a 9 kb transgenic *Vil1* promoter, allowing widespread recombination throughout the small (80%) and large intestine (40%) even 2 months after tamoxifen administration. This indicates that the artificial *Vil1* promoter is also active in most intestinal stem cells [23]. Thus, either differences in transgene design (conventional transgenics vs. knock-in) or in inducible systems (posttranslational estrogen ligand binding domain vs. destabilizing domain) might account for these findings.

In the next set of experiments, we assumed that similar to maturing enterocytes, *Vil1* would also be higher expressed in mature EECs than in their early precursors. Such a unique opportunity to specifically target mature EEC populations would open novel avenues in EEC research. That the villus EEC gene expression differs clearly from their early crypt-based precursors had been previously demonstrated by the usage of a fluorescent marker that changes its emission spectrum over time. Therefore, we thought labeling only mature villus EECs with our *Vil1-2A-DD-Dre* mouse strain in combination with existing Cre mice and Cre/Dre dependent tdTomato reporter might be an elegant proof-of-principle experiment. However, our premise that *Vil1* expression increases during EEC maturation and migration was invalid; in fact, scRNAseq revealed that L, D, and K cells express *Vil1* lower than I, N, and EC cells at any stage. Therefore, we were able to target only a minor population of I cells with high *Vil1* expression but not L and D cells in respective triple-transgenic mice. Our results that D cells are difficult to target with recombinase expression from one *Vil1* allele are in line with a recent publication successfully using *Vil1-p2a-FlpO* endogenous insertion with *Tac1-Cre*, *Npy1R-Cre, Gip-Cre, Gcg-Cre*, and *Cck-Cre* to efficiently express transgenes selectively in EC, K, L, and I lineages but not in D cells where the authors switched their strategy to *Vil1-Cre/Sst-ires-FlpO* combination [10]. In contrast, in another classical transgenic approach, where multiple transgenic copies drive the high expression of Flp recombinase from an artificial *Vil1* promoter, not only intestinal but also minor, rather neglectable intersectional labeling of pancreatic cells was observed [7]. Furthermore, the high and early Flp expression might also lower the stringency in targeting specific EEC cell types, e.g., the combination with *Gcg-Cre* labeled not only L cells but also *Pyy-*, *Cck-*, and *Gip*-expressing EEC cells [7]. Thus, in light of these and our findings, each intersectional toolbox to target specific intestinal EEC populations has its own benefits and limitations, but there is still space for further improvements. In fact, secretin (*Sct*, Figure 4i) driven recombinase expression (Dre or Flp) is a promising candidate for intersectional targeting of mature EECs since it is highly and specifically expressed in late EECs except for D cells.

## Figures and Tables

**Figure 1 cells-13-00102-f001:**
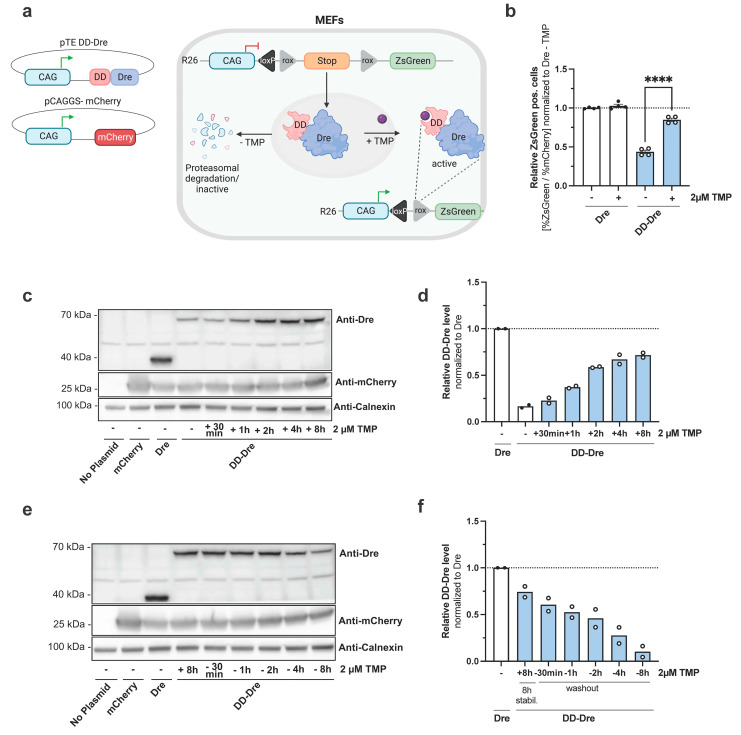
Generation of a TMP-inducible Dre system. (**a**) pTE-DD-Dre and pCAGGS-mCherry plasmids were transiently transfected into mouse embryonic fibroblasts carrying an *R26-rx-ZsGreen* reporter construct (MEF). Without trimethoprim (TMP), destabilizing domain (DD) targets DD-Dre for proteasomal degradation, whereas TMP treatment stabilizes DD-Dre, enabling excision of the rox-flanked STOP cassette of *R26-rx-ZsGreen* resulting in green fluorescence. (**b**) Quantification of ZsGreen^+^ cells normalized to mCherry^+^ cells (transfection control) of non-induced or TMP-treated MEFs transfected with the indicated plasmids. ZsGreen^+^/mCherry^+^ cells were quantified by FACS 48 h after transfection and normalized to pTE-Dre transfected cells w/o TMP. Gating strategy is shown in Appendix A. Statistical analysis was performed using Student’s *t*-test. Representative results of three independent experiments are shown (*n* = 4). (**c**) Western blot analysis using anti-Dre, anti-mCherry, and anti-Calnexin (loading control) antibodies of MEFs transiently transfected w/o plasmid or with pCAGGS-mCherry control or with pTE-Dre or pTE-DD-Dre treated with 1µM TMP for the indicated time spans to verify DD-Dre inducibility. Representative fluorescence microscopy images of co-transfected mCherry are shown in Appendix A. (*n* = 2). (**d**) Quantification of DD-Dre protein levels relative to Calnexin loading control in c) normalized to Dre protein level. (**e**) Western blot analysis using anti-Dre, anti-mCherry, and anti-Calnexin (loading control) antibodies of MEFs transiently transfected w/o plasmid or with pCAGGS-mcherry control with pTE-Dre or pTE-DD-Dre treated with 1 µM TMP for 8 h and consecutive TMP wash-out for indicated time points to verify DD-Dre instability. Representative fluorescence microscopy images of co-transfected mCherry are shown in Appendix A. (*n* = 2). (**f**) Quantification of DD-Dre protein relative to Calnexin loading control in (**e**) normalized to Dre protein level. Dashed lines represent respective normalization to one. Data are represented as mean ± SEM. **** *p* ≤ 0.0001.

**Figure 2 cells-13-00102-f002:**
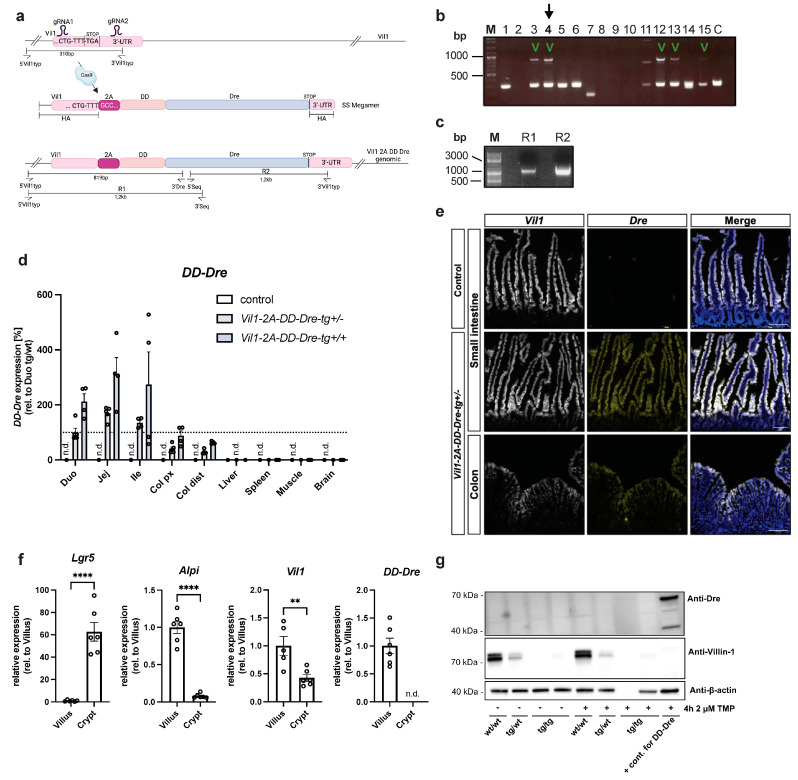
Generation of *Vil1-2A-DD-Dre* transgenic mice. (**a**) CRISPR/Cas9 strategy to insert ssMegamer encoding *2A-DD-Dre* into the stop codon of the endogenous *Vil1* gene. *Vil1-2A-DD-Dre* mice were generated by oocyte injection of C57BL/6N origin. gRNA1 and gRNA2 flank the stop codon of the *Vil1* gene, where, when co-injected with Cas9 protein, genomic DNA is cleaved. A ssMegamer repair template was co-injected that carried homology arms to the *Vil1* gene as well as an in-frame fusion of *2A-DD-Dre*, which, upon correct integration, resulted in *Vil1-2A-DD-Dre* genomic insertion. Genotyping was performed using external primers 5′Vil1typ with 3′Vil1typ and 3′Dre to result in a 310 bp wt band and an 819 bp insertion band. (**b**) Genotyping PCR of founder mice. The 5 animals that tested positive for 2A-DD-Dre insertion are marked. (**c**) Founder 4 was used to amplify 1.2 kb R1 and R2 PCR products (from (**a**) that were verified for correct integration by sequencing). (**d**) Female and male *Vil1-2A-DD-Dre*-tg+/− mice were intercrossed to obtain control, *Vil1-2A-DD-Dre-tg+/*−, and *Vil1-2A-DD-Dre-tg+/+* mice. DD-Dre expression in the duodenum, jejunum, ileum, proximal, and distal colon, as well as in liver, spleen, and muscle, and the brain was examined by qPCR derived from indicated mice (each circle one mouse). (**e**) Representative images of RNAScope ISH against *Vil1* and *Dre* mRNA on small intestinal and colon sections of indicated mice. Scale bar: 100 µm. (**f**) qPCR-based expression analysis of DD-Dre using mRNA derived from villus- or crypt-isolated IECs (each circle one mouse). Statistical analysis was performed using Student’s *t*-test. (**g**) Western blot analysis using anti-Dre, anti-VIL1, and anti-b-Actin (loading control) antibodies of lysates from IECs isolated from indicated mice that were non-induced or treated ex vivo with TMP for 4 h. Data are represented as mean ± SEM. ** *p* ≤ 0.01, **** *p* ≤ 0.0001.

**Figure 3 cells-13-00102-f003:**
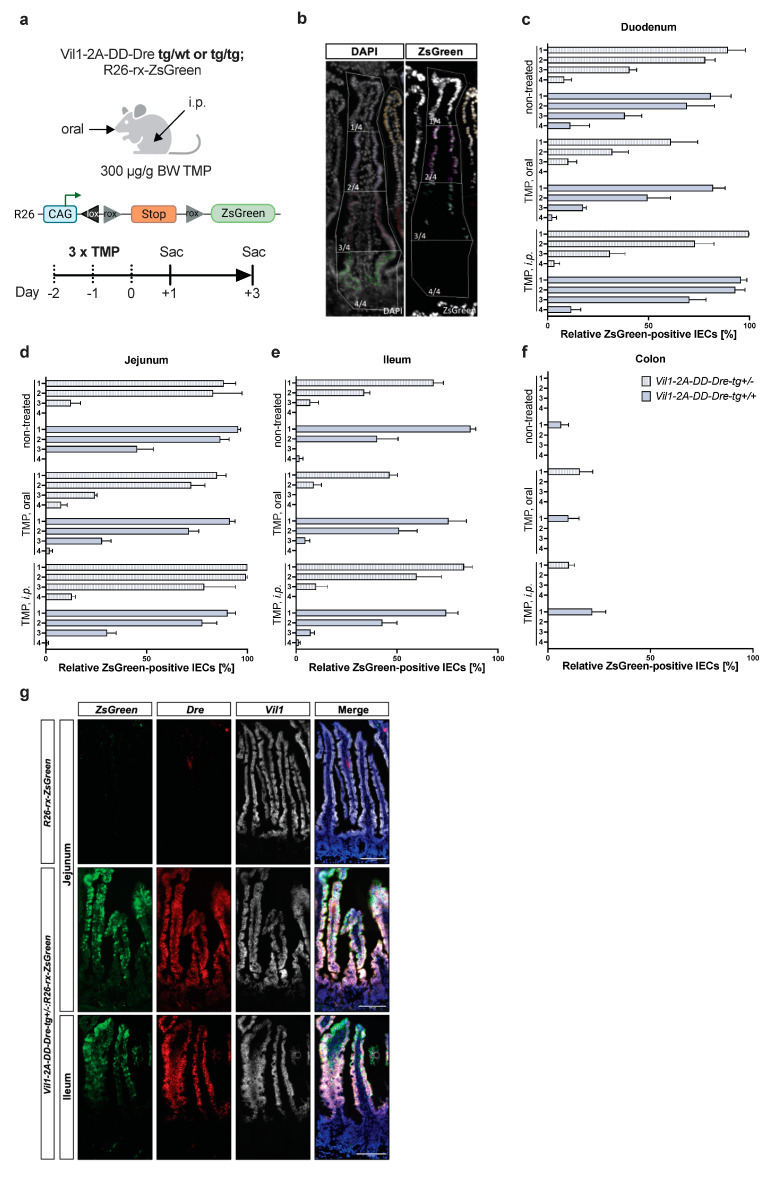
Efficient TMP-independent DD-Dre-mediated recombination of rox-flanked stop cassette in the small intestine of *Vil1-2A-DD-Dre* mice. (**a**) *Vil1-2A-DD-Dre-tg+/−*; or *tg+/+* were intercrossed with a *R26-rx-ZsGreen* reporter strain and either treated with 300 µg/g bodyweight TMP orally or by i.p. injection on 3 consecutive days. Controls did not receive TMP. Mice were sacrificed 1–3 days after the last TMP administration to isolate intestinal samples and quantify DD-Dre activity. (**b**) Representative image of intestinal ZsGreen fluorescence counterstained with DAPI. The crypt–villus axes were divided into 4 quadrants from apical to basal to quantify percentage of ZsGreen^+^ positive cells in (**c**) duodenum, (**d**) jejunum, (**e**) ileum, and (**f**) colon of indicated mice; light blue heterozygous, dark blue homozygous *Vil1-2A-DD-Dre-tg* mice. (**g**) Representative images of multiplex RNAScope ISH of jejunal and ileal sections of indicated mice using probes to detect *ZsGreen*, *Dre*, and *Vil1* mRNA. Scale bars: 100 µm.

**Figure 4 cells-13-00102-f004:**
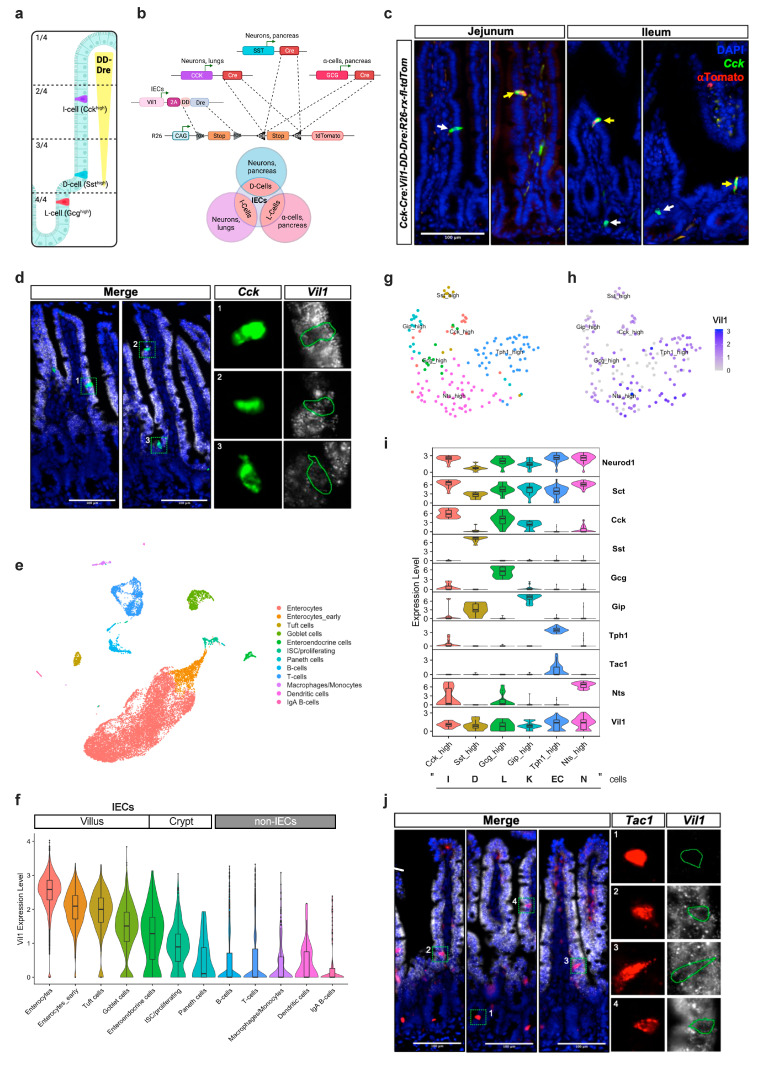
An intersectional approach to target mature EEC populations. (**a**) Scheme of intestinal distribution of *Cck* expressing I cells, *Sst* expressing D cells, and *Gcg* expressing L cells along the crypt–villus quadrants. I cells can be found either near the crypt–villus junction or higher up the villus axis. (**b**) Intersectional strategy using *Vil1-2A-DD-Dre*; *R26-rx-fl-tdTomato* with combinations of *Cck-Cre*, *Sst-Cre,* and *Gcg-Cre* mice to target specifically mature I, D, and L cells, respectively. Note that *DD-Dre* is expressed only in mature IECs, whereas *Cck-Cre* is expressed in I cells, neurons, and lungs; *Sst-Cre* is expressed in D cells, neurons, and pancreas; *Gcg-Cre* is expressed in L cells and pancreatic alpha cells. (**c**) Anti-tdTomato IHC combined with RNAScope ISH against *Cck* counterstained with DAPI of jejunum and ileum of indicated mice. Yellow arrows indicate *Cck*/tdTom double-positive cells. White arrows indicate *Cck*-only positive cells. (**d**) Multiplex RNAscope ISH of *Vil1* and *Cck* counterstained with DAPI revealed lower *Vil1* expression in I cells than in neighboring enterocytes. Scale bars: 100 µm. (**e**) UMAP graph of single-cell transcriptomics of alive small intestinal cells of 12-week-old mice (3 mice were pooled). (**f**) Violin plot for *Vil1* expression across the indicated cell clusters. (**g**) Supervised re-clustering of EEC types according to their hormonal expression. (**h**) *Vil1* expression in EEC subtypes. (**i**) Violin plots showing expression levels of EEC subtype markers and *Vil1* in the defined EEC populations from (**g**). (**j**) Multiplex RNAscope ISH of *Vil1* and *Tac1* counterstained with DAPI confirmed *Vil1* expression.

## Data Availability

Single-cell RNA sequencing data were deposited in NCBI Gene Expression Omnibus with an identifier: GSE250433.

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
