# Peer review of "An Approach to Intersectionally Target Mature Enteroendocrine Cells in the Small Intestine of Mice"

_cells, 2024, doi:10.3390/cells13010102_

Round 1
Reviewer 1 Report
Comments and Suggestions for Authors
In this article the authors describe their attempt to create an inducible genetic intersectional model to target enteroendocrine cells (EEC) in the intestinal epithelium. EEC subtype-specificity was aimed to be achieved using widely available hormone specific Cre-driver lines. Intestinal specificity was aimed at by creation of a new transgenic mouse model, expressing a destabilised DD-Dre-recombinase under the control of the Villin-1 promoter, which was hoped to enable time defined Dre-recombination dependent on the timed addition of a stabilizer (TMP) preventing protein degradation due to the DD-signal – the stabilization principal was confirmed with heterologous expression analysis in an in vitro system. Reporter dependent on either or both Cre- and Dre-recombination were employed. Unfortunately, the Vil1-2A-DD-Dre knock-in strategy did not work, as the 2A sequence failed to produce a bicistronic mRNA and instead an unstable monocistronic fusion protein was expressed. Dre- and Villin-mRNA were only detectable in the more mature sections along the crypt-villus axis in the small intestine and hardly at all in the large intestine. This is surprising, as Vil1-Cre mice are widely used to activate or to knock-out alleles in all epithelial cells, consistent with sufficient Cre-activity in intestinal stem cells located at the bottom of crypts in small and large intestine. Presumably the DD-approach restricted activity in this niche.
The discussion focuses on co-expression of both Cre- and Dre-recombinase to activate flox/rox-reporters, emphasising details of lower villin expression in some EEC subtypes. However, to my understanding the removal of the flox and the rox STOP cassettes do not need to happen at the same time. This likely explains why the previous two publications (Ref1, Bai et al and Ref13 Hayashi et al.) using a similar genetically intersectional approach employing Vil1-FlpO mice were more successful in targeting EEC-subtypes, likely the result of Vil1-driven STOP-removal in the stem cell and removal of the prevailing STOP at a later time when specific hormone expression is switched on at a sufficient level to drive Cre-dependent recombination. Given that this lineage tracing could also work the other way round, cells that previously had switched on Gcg-Cre and mature along the crypt-villus axis will start to express reporters if Vil1-DD-Dre activity becomes sufficiently high; I assume this would limit the inducibility aspect of the attempted mouse, even if it would have worked – maybe the generally low expression of Vil1-DD-Dre in EECs will prevent such cells to report and this is likely, as the authors suggest, the reason for the limited labelling even of CCK-expressing I-cells. Regardless, the new model described here seems of very limited use, due to an unfortunate design-fault. The presented data will nonetheless be interesting to the specialist reader – e.g. villin1 mRNA and protein expression details.
Specific comments;
1) The abstract is misleading, as it does not indicate that the new DD-Dre mouse had a design fault and implies that Dre-expression per se would not work to label D and L-cells, suggesting that this is likely due to Villin-expression profiles. Please rewrite to better reflect the findings.
2) Fig1b shows ~45% rox-recombination activity of DD-DRE in the absence of TMP. I would consider this to be problematic as this would indicate that, provided sufficient expression levels of DD-DRE would have been achieved, the system would be constitutively active rather than inducible. This could be discussed.
3) Was expression of Vil1-p2A-DD-DRE in the pancreas tested – in recent publications using a similar genetically intersectional strategy creating Vil1-FlpO mice some low, but detectable expression in the pancreas was present. Please assess and discuss.
4) I don’t agree with the conclusion of complete absence of Villin in homozygous mice as, admittedly very faint, there is nonetheless a band in Fig2g lines 4 and 8. This could be mentioned.
5) Given that the in vitro kinetics indicated wash out of TMP stabilisation within 8 hours I have concerns using 1-3 days of TMP treatment to time dependently label cells. As the DD-part did not work and some Dre reporting was observed in the villi sections regardless of TMP, this is of minor importance for the current manuscript. However, given the relatively fast turn-over of the small intestinal epithelium, 3 days should label >50% of all epithelium if the recombination would occur in the intestinal stem cell – this should be taken into account should the authors redesign their inducible strategy using a p2A rather than a 2A bicystronic sequence and could be discussed.
6) “However, the low abundant EECs, constituting approximately 1% of the intestinal epithelium, do not jump onto this conveyor belt of moving IECs, but instead stay close to the crypt-villus junction (Gehart et al., 2019; Latorre et al., 2016).” –1% is approximately correct for all EECs. However, whilst a minority of EECs appears to undergo slower turnover, EECs do move from the crypt to the villus tip. Please reformulate to reflect this. {In this context, the authors seemed surprised to “only” observe 167 EECs in their 10x single cell analysis, but this is in line with ~1%}.
7) “To establish an inducible Dre system, we generated a plasmid where expression of a N-terminally fused destabilizing domain (DD) to Dre is regulated by the CAG promoter…” – grammatically not quite right; please reformulate: expression of the fusion protein is regulated, not expression of the DD.
Comments on the Quality of English LanguageVery good - two minor points highlighted above.
Reviewer 2 Report
Comments and Suggestions for Authors
The manuscript by Vossen et al. presents interesting data. The manuscript is well written and detailed methods are given. With regards to scientific progress and 3Rs it is very important that also “unexpected” outcomes are published.
Major comments:
It is well known, that Vil-Cre (ERT2) is able to also target stem cells and colonic cells. Thus, even though Villin1 expression might be lower in some cell subtypes, there might be other reasons why these cells are not affected in the system presented. This might be due differences in proteasome activity or other differences in distinct cell types and should be at least discussed if not tackled experimentally.
No Villin1 expression is given in crypt versus villus (figure 2f) -> this should be included as it is discussed as main reason for the observations.
Figure 2d -> what about kidney or lung, tissues that are likely to express villin1?
Figure 1b: a “not-even-2-fold” increase is rather low upon induction. The presented “relative positive cells” do not allow to judge on leakage of the system.
The authors state that that “the intersectional genetic approach described here applicable for the investigation of mature EECs”. This statement is only partly correct as it is not desirable to use a Vil1-ko model to study specific effects (even though no macroscopic aberrations are observed).
To this end, the presented model did not meet expectations. However, there are several interesting findings. This should be discussed more clearly and honestly in the manuscript.
in line 81 a "s" is missing (subpopulations)
Reviewer 3 Report
Comments and Suggestions for Authors
Vossen and collaborators present some original work in a clear an well written paper in which they tried to develop a new strategy to specifically target mature enteroendocrine cells (EECs) using intersectional genetic approach. They developed a Vil1-2A-DD-Dre mouse model which enable Dre activity only in cells expressing high levels of Vil1 to be crossed with EEC specific cre expressing mice to label specifically mature EECs based on the hypothesis of mature intestinal epithelial cells expressing high levels of Vil1.
The aim to target mature EECs to discriminate their role from early EECs is of interest as these cells change their identity as their mature, however, the results described using this strategy show that this approach might not be optimal due to the low Vil1 expression in some EEC subpopulations. Nevertheless, this work is of interest for the community as it clearly shows the limitation of Vil1 marker to target all intestinal epithelial cells. Before publication, I do have some remarks and questions, hoping to improve the already good quality of this paper:
- The message about Vil1 expression in IELs is not clear and there is some confusion between no expression and absence of detection or low levels of expression. Indeed, as shown in figure 4f, all IELs express Vil1, but at different levels between cell populations. Line 450-453 should be reformulated to indicate no detectable or sufficient expression. Moreover, in figure 2F, expression of Vil1 in isolated crypts and vill should be also shown. Relative gene expression should be shown in a log scale to represent in a similar way increased and decreased expression.
- Mature EECs have been associated with increased expression of secretin levels (Beumer 2018, Billing 2019) in all cells except D cells, have the authors looked at secretin expression in correlation with Vil1 in their data (figure 4)? Could they also comment on the possibility to use secretin as a specific mature EEC marker? The hypothesis about Tac1
- In figure 3, different conditions seem to be used with some mice injected ip while others received TMP by gavage and time of analysis after last injection varied. Can the authors clarify what they represent and if any difference between treatments were observed? Moreover, the non TMP treated mice results should be shown in a similar way (% of positive IECs) in all different regions as it is a control for TMP treatment, especially if the authors claim that TMP administration does not change anything.
- To validate the absence of effect of TMP treatment in vivo and rule out the hypothesis that TMP does not reach IELs, especially distal ones, can the authors generate organoids from different distal regions to test in vitro their system with a better model of IELs than MEFs?
- In the introduction, authors claim that EECs do not jump in the conveyor belt and stay in the crypt villus junction. This is highly questionable and several data, including own authors’ results in this paper, clearly show that EECs can be scattered all along the crypt-villus axis.
- In figure 4i, scale should be changed to avoid violin plots to reach the top of the graphs and in figure 1, number of replicates are lacking.
Round 2
Reviewer 2 Report
Comments and Suggestions for Authors
Thank you for the clear presentation of changes and response.